# A process-oriented evaluation of CAMS reanalysis ozone during tropopause folds over Europe for the period 2003–2018

Dimitris Akritidis[1], Andrea Pozzer[2], Johannes Flemming[3], Antje Inness[3], Philippe Nédélec[4], and Prodromos Zanis[1]

[1]Department of Meteorology and Climatology, School of Geology, Aristotle University of Thessaloniki, Thessaloniki, Greece
[2]Atmospheric Chemistry Department, Max Planck Institute for Chemistry, Mainz, Germany
[3]European Centre for Medium-Range Weather Forecasts (ECMWF), Reading, UK
[4]LAERO, Université de Toulouse, UT3, CNRS, IRD, Toulouse, France

**Correspondence:** Dimitris Akritidis (dakritid@geo.auth.gr) and Andrea Pozzer (andrea.pozzer@mpic.de)

**Abstract.** Tropopause folds are the key process underlying stratosphere-to-troposphere transport (STT) of ozone, thus, affecting tropospheric ozone levels and variability. In the present study we perform a process-oriented evaluation of Copernicus Atmosphere Monitoring Service (CAMS) reanalysis (CAMSRA) $O_3$ during folding events, over Europe and for the time period from 2003 to 2018. A 3-D labeling algorithm is applied to detect tropopause folds in CAMSRA, while ozonesonde data from
WOUDC (World Ozone and Ultraviolet Radiation Data Centre) and aircraft measurements from IAGOS (In-service Aircraft for a Global Observing System) are used for CAMSRA $O_3$ evaluation. The profiles of observed and CAMSRA $O_3$ concentrations indicate that CAMSRA reproduces the observed $O_3$ increases in the troposphere during the examined folding events. Nevertheless, at most of the examined sites, CAMSRA overestimates the observed $O_3$ concentrations, mostly at the upper portion of the observed increases, with a median fractional gross error (FGE) among the examined sites > 0.2 above 400 hPa.
The use of a control run without data assimilation, reveals that the aforementioned overestimation of CAMSRA $O_3$ arises from the data assimilation implementation. Overall, although data assimilation assists CAMSRA $O_3$ to follow the observed $O_3$ enhancements in the troposphere during the STT events, it introduces biases in the upper troposphere resulting in no clear quantitative improvement compared to the control run without data assimilation. Less biased assimilated $O_3$ products, with finer vertical resolution in the troposphere, in addition to higher IFS (Integrated Forecasting System) vertical resolution, are
expected to provide a better representation of $O_3$ variability during tropopause folds.

## 1   Introduction

Ozone has multiple roles in the Earth's troposphere, making it one of the most important trace gases. It is a major source of the OH radical which controls the oxidizing capacity of the troposphere, and it is also a short-lived climate forcer being an important greenhouse gas, especially in the upper troposphere and lower stratosphere. Near the surface ozone is a pollutant
detrimental to human health, crops, and ecosystems (Monks et al., 2015; Young et al., 2018). The Intergovernmental Panel on Climate Change (IPCC) Sixth Assessment Report (AR6) assessed that tropospheric ozone has increased since the mid-20th century by 30–70% across the Northern Hemisphere based on sparse historical surface/low altitude data (Gulev et al.,

2021; Szopa et al., 2021). The tropospheric ozone budget is controlled by chemical production and loss, by stratosphere-troposphere exchange (STE), and by deposition at the Earth's surface, whose magnitude can vary widely across chemistry climate models (Young et al., 2018; Griffiths et al., 2021; Szopa et al., 2021). The net stratospheric influx results from STE processes, comprised of stratosphere-to-troposphere transport (STT) and troposphere-to-stratosphere transport (TST) (Stohl et al., 2003). The latter constitutes an important pathway through which very short lived substances (VSLS), emitted at the surface, can be transported to the lower stratosphere influencing ozone (Levine et al., 2007; Aschmann et al., 2009; Liang et al., 2014). The main mechanism for STT is tropopause folding (Stohl et al., 2003), which results in the downward transport of stratospheric ozone-rich air into the troposphere, a process known as stratospheric intrusion (Danielsen and Mohnen, 1977). Therefore, tropopause folding events affect tropospheric composition and in particular tropospheric ozone levels (Beekmann et al., 1997; Ott et al., 2016; Cooper et al., 2005; Tarasick et al., 2019; Zhao et al., 2021a), especially in regions that are known to be hot spots of fold activity (Zanis et al., 2014; Akritidis et al., 2016; Ojha et al., 2017). Occasionally, during deep and intense folding events, stratospheric air is transported down to the lower troposphere or even to the planetary boundary layer leading to changes in tropospheric and surface ozone concentrations (Langford et al., 2009; Lin et al., 2015; Knowland et al., 2017). Model projections suggested that under a changing climate tropopause folds will be associated with both future increases and interannual variability in ozone STT (Akritidis et al., 2019).

The spatial and temporal characteristics of tropopause folds occurrence around the globe have been the subject of study in recent years, suggesting the jet stream location, intensity, and seasonality as their main drivers (Elbern et al., 1998; Sprenger et al., 2003; Škerlak et al., 2014; Akritidis et al., 2021). The springtime western United States region is a hot spot of deep folding events with well-known implications for tropospheric ozone and air quality (Langford et al., 2009; Lin et al., 2012, 2015; Knowland et al., 2017). Recently, Luo et al. (2019) explored the seasonal features of tropopause folds over the Tibetan Plateau where folds occur frequently (Tyrlis et al., 2014), while other studies investigated the effect of tropopause folds on lower tropospheric ozone levels and air quality in China (Lu et al., 2019; Zhao et al., 2021b, a). Regarding the broader European region, the summertime Eastern Mediterranean is a well-known hot spot of fold activity (Tyrlis et al., 2014; Akritidis et al., 2016), resulting from the interaction of the subtropical jet stream and the South Asian Monsoon (Tyrlis et al., 2014), while further north folds occur in the vicinity of cyclones (Reutter et al., 2015; Antonescu et al., 2013; Knowland et al., 2015). During the past two decades several studies have explored the impact of stratospheric intrusions on tropospheric ozone levels and variability over Europe (Stohl et al., 2000; Cristofanelli et al., 2006; Trickl et al., 2020), as well as the quality of their forecast (Zanis et al., 2003; Trickl et al., 2010; Akritidis et al., 2018).

Nowadays, a comprehensive framework to study the contribution of stratospheric intrusions to tropospheric ozone are atmospheric composition reanalysis products that provide global meteorological and ozone data in relatively high spatial and temporal resolution. Yet, before estimating the impact of STT events on tropospheric ozone, a process-oriented evaluation of the reanalysis product during such events is deemed essential. The latest reanalysis of atmospheric composition produced by the European Centre for Medium-Range Weather Forecasts (ECMWF) is the Copernicus Atmosphere Monitoring Service (CAMS) reanalysis (CAMSRA) (Inness et al., 2019). Within the framework of the CAMS service element CAMS_84, Akritidis et al. (2018) evaluated the performance of the ECMWF Integrated Forecasting System (IFS) in forecasting the observed $O_3$

increases in the troposphere during a deep STT event over Europe. However, apart from such individual case studies, there is no long-term evaluation of IFS during STT events. Recently, Akritidis et al. (2021) using a fold detection algorithm constructed a global record of tropopause folds in CAMSRA for the period from 2003 to 2018.

In the present study we perform a process-oriented evaluation of CAMSRA $O_3$ during STT events selected from the CAMSRA tropopause folds database by Akritidis et al. (2021), for the European region and over the time period 2003–2018. Compared with other regions worldwide, the European region exhibits relatively higher observational data availability for the examined period. In addition, the role of IFS chemical data assimilation in $O_3$ STT is explored. Section 2 describes WOUDC (World Ozone and Ultraviolet Radiation Data Centre) and IAGOS (In-service Aircraft for a Global Observing System) $O_3$ data used for the evaluation; ECMWF IFS system and CAMSRA data; the 3-D labeling algorithm applied for tropopause fold-detection; and the methodological approach for the selection of STT events. Section 3 presents the main evaluation results, and finally Section 4 summarises the key findings of the study.

## 2 Data and Methodology

### 2.1 Observational data

To evaluate CAMSRA $O_3$ during folding events, ozonesonde measurements were obtained from the WOUDC network (WMO/GAW Ozone Monitoring Community) (last access: 19 April 2020) for nine European sites, namely Lerwick, United Kingdom (UK) (LER); Uccle, Belgium (UCC); Hohenpeissenberg, Germany (HOH); Payerne, Switzerland (PAY); Legionowo, Poland (LEG); Madrid, Spain (MAD); De Bilt, the Netherlands (DBI); Lindenberg, Germany (LIN); Prague, Czech Republic (PRA). At all sites the measurements are carried out with electrochemical concentration cell (ECC) ozonesondes (Komhyr, 1969), except at Hohenpeissenberg where the Brewer Mast ozonesonde (Brewer and Milford, 1960) is used. Both ozonesonde types are based on the same measurement principle of ozone electrochemical detection in potassium iodine. The major differences between ECC and Brewer Mast ozonesondes are that the latter uses only one reaction chamber, and a silver anode instead of a platinum anode, requiring an external electrical potential in contrast to the ECC (Beekmann et al., 1994). The precision of ECC ozonesondes in the troposphere (below 200 hPa) is between -7% and +17%, as reported by Komhyr et al. (1995), while for the Brewer Mast ozonesondes the same order of precision was found by Steinbrecht et al. (1998). The ozonesonde observations are compared against CAMSRA $O_3$ concentrations of the nearest grid point and timestep.

Aircraft ozone measurements from the IAGOS (In-service Aircraft for a Global Observing System) programme were also used (last access: 21 April 2020). Within the framework of IAGOS, instruments are carried on commercial airlines, measuring ozone, carbon monoxide and water vapour along with meteorological parameters and cloud particles. Details of the IAGOS project can be found in Petzold et al. (2015), with the technical details of the instrumentation, operations, and validation presented in Nédélec et al. (2015). Five IAGOS airports were selected for the evaluation, based on their data temporal coverage, namely Paris, France (PAR); Düsseldorf, Germany (DUS); Frankfurt, Germany (FRA); Munich, Germany (MUN); Vienna, Austria (VIE). The IAGOS $O_3$ data have an accuracy of $\pm$ 2 ppb, a precision of $\pm$ 2%, and a detection limit of 2 ppb (Blot et al., 2021). Landing and take-off $O_3$ profiles are compared against CAMSRA $O_3$ profiles. It should be noted that the IAGOS

profiles are not strictly vertical. To this end, and in order to perform a more realistic evaluation of CAMSRA O$_3$, according to the flight position (longitude, latitude, pressure) the respective CAMSRA grid points are extracted at the nearest time to that of the take-off or landing. The selection of both ozonesonde sites and IAGOS airports was based in the availability of at least 500 profile observations throughout the 2003-2018 period. This objective criterion ensures a sufficient number of both observational

sites and folding events to be selected for the analysis. It is noteworthy to mention that both ozonesondes and IAGOS profiles are not assimilated and hence they constitute completely independent validation data. The location of the examined WOUDC ozonesonde sites and IAGOS airports are depicted in Figure 1.

## 2.2 CAMS reanalysis

CAMSRA is the latest reanalysis dataset produced by ECMWF, including 3-dimensional fields of meteorological, chemical,

and aerosol species for the period from 2003 onwards. It comes as a follow-up of the previous successful reanalysis products, the Monitoring Atmospheric Composition and Climate (MACC) reanalysis (MACCRA) (Inness et al., 2013) and the CAMS interim reanalysis (CAMSIRA) (Flemming et al., 2017). CAMSRA is based on the ECMWF's IFS CY42R1 cycle and the 4D-VAR data assimilation system (Inness et al., 2019). In more detail, it is based on the minimization of a penalty function that takes the deviations of the model's background fields from the observations to provide the optimal forecast during 12-hour

assimilation windows (from 09 UTC to 21 UTC and 21 UTC to 09 UTC) by modifying accordingly the initial conditions. To this end, satellite retrievals of total column CO, tropospheric column NO$_2$, aerosol optical depth and total column, partial column and profile ozone retrievals are assimilated in the IFS system. More details on the satellite retrievals (product, satellite, period) assimilated in CAMSRA can be found in Table 1 of the CAMSRA evaluation study by Wagner et al. (2021). In addition, meteorological observations, including satellite, PILOT, in situ, radiosonde, dropsonde, and aircraft measurements are

also incorporated in IFS. The chemical mechanism used in the IFS is an extended version of the Carbon Bond 2005 (CB05) tropospheric chemical mechanism (Flemming et al., 2015) and stratospheric ozone chemistry is parameterised by a "Cariolle-scheme" (Cariolle and Déqué, 1986; Cariolle and Teyssèdre, 2007). The emissions consist of the MACCity (MACC and CityZEN EU projects) anthropogenic emissions (Granier et al., 2011), the GFAS (Global Fire Assimilation System) fire emissions (Kaiser et al., 2012), and the MEGAN2.1 (Model of Emissions of Gases and Aerosols from Nature) biogenic emissions

(Guenther et al., 2006). The CAMSRA data have a spatial resolution of approximately 80 km (0.7$^o$ x 0.7$^o$ grid) with 60 hybrid sigma/pressure (model) levels (13 levels between approximately 400 and 100 hPa) in the vertical (top level at 0.1 hPa), and a temporal resolution of 3 hours. The quality of the CAMSRA O$_3$ field is documented in Wagner et al. (2021) and comprehensive validation reports that can be found on the CAMS website https://atmosphere.copernicus.eu/eqa-reports-global-services (e.g. Errera et al., 2021).

To investigate the role of chemical data assimilation in tropospheric ozone representation during folding events, a control simulation of IFS without the use of chemical data assimilation (CAMSRA no DA) is also used. As it would have been computationally too expensive to produce a control analysis experiment that was identical to CAMSRA but did not actively assimilate observations of reactive gases, a forecast run was carried out that applied the same settings (model code, resolution, emissions) as used in CAMSRA. The control run was carried out as a sequence of 24 hours. The meteorological initial conditions were

taken from CAMSRA, but the initial conditions for the atmospheric composition species, including ozone, from the previous forecast. It thus allows us to detect the impact of the assimilation of e.g. ozone data by comparing its ozone fields with CAMSRA. This approach was also followed by Akritidis et al. (2018) in their evaluation of CAMS forecasting systems during a deep STT event over Europe, indicating an overall improvement of IFS performance due to the chemical data assimilation implementation.

Apart from $O_3$, a stratospheric ozone tracer ($O_{3s}$) is also used from CAMSRA providing a diagnostic of $O_3$ STT. In principal, $O_{3s}$ in IFS is defined identically with $O_3$ in the stratosphere, yet, in CAMSRA $O_{3s}$ is equal to the modeled (Cariolle scheme) $O_3$ tracer and not the assimilation-resulted $O_3$. In the troposphere $O_{3s}$ is subject to transport and chemical destruction just like $O_3$. The tropopause in CAMSRA is calculated based on the temperature lapse rate, switching the chemistry scheme from CB05 (troposphere) to Cariolle (stratosphere) accordingly. It should be noted, that $O_{3s}$ is used here only as a qualitative diagnostic of 135 ozone STT, to support evidence of stratospheric ozone downward transport during the folding events.

## 2.3 Fold detection in CAMS reanalysis

Tropopause folds are identified in CAMSRA using the latest version of the 3-D labeling and fold detection algorithm by Škerlak et al. (2015), initially developed by Sprenger et al. (2003). Here, we adopted the 3-D labeling algorithm to detect folds in CAMSRA, using as inputs the fields of potential vorticity (PV), potential temperature, specific humidity, and surface pressure. 140 The 3-D fields of pressure are constructed and the pressure level of the dynamical tropopause (Holton et al., 1995; Stohl et al., 2003) is determined using the lower of the isosurfaces of PV at 2 PVU and potential temperature at 380 K. Subsequently, the vertical profile for each grid point is examined and a fold is assigned when multiple crossings of the tropopause are identified. Still, there are specific cases where air with PV > 2 PVU is either not connected to the stratosphere (stratospheric cut-offs) or is not of stratospheric origin (diabatic PV anomalies or surface-bound PV anomalies) which should not be considered 145 as stratospheric. To this end, the 3-D labeling algorithm, using physical and geometrical criteria, labels the air masses as follows: tropospheric (label=1); stratospheric (label=2); stratospheric cut-off or diabatically produced PV anomaly (label=3); tropospheric cut-off (label=4); surface-bound PV anomaly (label=5). The diabatically produced PV anomalies merged with the stratosphere are distinguished using a specific humidity threshold of 0.1 $g\,kg^{-1}$. Further details on the criteria used for the 3-D labeling can be found in Škerlak et al. (2015). Therefore, a fold is identified when a $2 \to 1 \to 2 \to 1$ or 3 transition is 150 detected on a vertical profile (from top to bottom), with the algorithm outputting a binary variable (0:no fold, 1:fold) for every grid point and time step. In addition, the upper ($p_u$), middle ($p_m$), and lower ($p_l$) pressure levels of the tropopause crossings are identified along with the difference $\Delta p = p_m - p_u$, which depicts the vertical extent of the fold. The spatial distribution of CAMSRA monthly mean tropopause folds (with $\Delta p \geq 50$ hPa) frequency over Europe for the period 2003–2018 is presented in Figure S1 of the Supplement.

155 ## 2.4 Selection of STT events

To perform the process-oriented evaluation of CAMSRA $O_3$, the STT events are selected for each WOUDC ozonesonde site applying the following methodology:

(a) For every ozonesonde profile the time and location of release are extracted.

(b) For the CAMSRA grid cell including the ozonesonde site location and for the CAMSRA 3-hour timesteps before and after the time of ozonesonde release, the presence of a tropopause fold with $\Delta p \geq 50$ hPa is explored (e.g. if the ozonesonde release was 14:00 UTC, we search for folds at the 12:00 UTC and 15:00 UTC timesteps of the respective grid cell).

(c) If a fold is found the respective ozonesonde profile is classified in "STT events", while otherwise is classified in "rest of events".

(d) The ozonesonde data are vertically interpolated (linear) with a step of 25 hPa, and only the profiles exhibiting a data completeness $\geq 75$ % from 900 to 300 hPa are kept in the STT events and rest of events records. Merging STT events with the rest of events provides the climatology of ozonesonde profile.

The same approach is followed (steps a, c, and d) for the IAGOS data with one difference in step b. Since the aircraft measurement profiles during take-off and landing are not strictly vertical, a tracking of the aircraft position is performed and the respective CAMSRA grid cells that include the aircraft route are extracted. Subsequently, the presence of a tropopause fold with $\Delta p \geq 50$ hPa is explored if it is found in at least one of the extracted grid cells. A schematic representation of the applied methodology for the STT events selection is illustrated in Figure 2. For direct comparison with observations, CAMSRA $O_3$ concentrations are also vertically interpolated (linear) with a step of 25 hPa.

## 3 Results

### 3.1 Comparison of observed and CAMSRA climatological $O_3$ profiles

Before proceeding with the process-oriented evaluation of CAMSRA $O_3$ during the STT events, we present a comparison of CAMSRA $O_3$ profiles against observations (ozonesonde and aircraft measurements) during all the events (STT events + rest of events), to ensure that CAMSRA reproduces the climatological features of the observed $O_3$ profiles at the examined European sites. Figure 3 presents the climatological $O_3$ profiles of both observations and CAMSRA for all examined WOUDC and IAGOS sites. As depicted, CAMSRA captures the features of observed vertical $O_3$ profiles, with a common characteristic at all sites being an overestimation of CAMSRA mostly in the upper troposphere, which is also seen in the evaluation studies by Inness et al. (2019); Huijnen et al. (2020); Wagner et al. (2021). More specifically, CAMSRA exhibits higher $O_3$ concentrations throughout the troposphere at Hohenpeissenberg and Paris, and to a lesser extent at Frankfurt and Munich, with the greatest overestimation seen in the upper troposphere at all sites. Similar $O_3$ overestimations in the free troposphere over Hohenpeissenberg are also reported for previous ECMWF atmospheric composition reanalysis products such as the MACC reanalysis (see Figures 7 and 8 in the evaluation study by Katragkou et al. (2015)). At Payerne and Düsseldorf, CAMSRA $O_3$ is only overestimated above 500 hPa. At the rest of the sites CAMSRA $O_3$ is quantitatively in very good agreement with observations. The differences seen in the comparison between the observed and CAMSRA $O_3$ concentrations among the examined sites are presumably related to regional differences and uncertainties in $O_3$ precursor emissions affecting modeled local net

photochemical $O_3$ production rates, as well as the spatiotemporal representativeness of WOUDC vertical profiles and IAGOS aircraft take-off/landing routes by the selected CAMSRA grid points and time steps.

## 3.2 Evaluation of CAMSRA $O_3$ during STT events

In Figures 4 and 5 we present CAMSRA and observed $O_3$ profiles averaged during the selected STT events and the rest of events at the WOUDC ozonesonde sites and IAGOS airports, respectively. Also shown are the respective CAMSRA $O_{3s}$
profiles. As expected, both CAMSRA and observations exhibit higher $O_3$ concentrations in the middle and upper troposphere for the STT events compared to the rest of events, at all examined sites. Similarly, CAMSRA $O_{3s}$ concentrations for STT events are higher than those for the rest of events, resembling the respective CAMSRA $O_3$ enhancements in the troposphere. This highlights the stratospheric contribution in $O_3$ increases during the selected tropopause folding episodes. Overall, CAMSRA $O_3$ is in a satisfactory agreement with the observed $O_3$ enhancements in the troposphere during the STT events, still exhibiting
specific limitations. A feature seen in some observational sites (Uccle, Hohenpeissenberg, Legionowo, Prague, Paris, and Düsseldorf), is that although CAMSRA follows the observed $O_3$ increases in the troposphere it misses the observed decrease back to normal tropospheric $O_3$ values, resulting in overestimation of $O_3$ in the upper troposphere. As mentioned above, the $O_3$ overestimation in the upper troposphere is an already known issue in both CAMS near-real-time analysis and reanalysis products. This might be due to a bias in some of the assimilated data, the likely insufficient vertical resolution of $O_3$ data
assimilated (total column and stratospheric profiles) in IFS to capture STT events, and the $O_3$ background error formulation in data assimilation. In particular, CAMSRA $O_3$ vertical profiles during both STT and rest of events exhibit a better agreement in the upper troposphere with observations during the years 2003 and 2004, indicating that the inclusion of the Aura data in the assimilation system from August 2004 and on is likely to result in $O_3$ overestimation in the upper troposphere (Figures S2 and S3 in the Supplement). In addition, the individual dynamics, and the different vertical location and geometrical characteristics
of the selected STT events, especially for observational sites with not so extended number of events, may form somehow unique structures of CAMSRA and observed O3 deviations. In particular, for sites exhibiting a very small number of STT events over the years (e.g. Legionowo), the $O_3$ vertical variability is not smoothed out and CAMSRA is not found able to reproduce the high resolution features of $O_3$ increase, probably due to its coarser vertical resolution compared to ozonesonde measurements.

For a quantitative comparison between CAMSRA and observations, we present in Figure 6 the vertical profiles of fractional
gross error (FGE) and modified normalized mean bias (MNMB) of CAMSRA $O_3$ for the WOUDC ozonesonde sites and IAGOS airports. The FGE is a normalized version of the mean error, while the MNMB is a normalized version of the mean bias. Both metrics are normalized by the mean of the observed and model (here CAMSRA) values, being dimensionless and relative, thus suitable to use at different heights in the troposphere. FGE and MNMB are insensitive to outliers in the distribution, and range between 0 to 2 and -2 to 2, respectively, behaving symmetrically with respect to under- and overestimation:

$$ FGE = \frac{2}{N} \sum_{i}^{N} \left| \frac{M_i - O_i}{M_i + O_i} \right| \tag{1} $$

$$MNMB = \frac{2}{N} \sum_{i}^{N} \frac{M_i - O_i}{M_i + O_i} \qquad (2)$$

where $M_i$ is the model (CAMSRA here) value for the $i$th STT event, $O_i$ is the corresponding observed value, and N is the number of STT events. As can be seen in Figure 6a the FGE is mostly increasing with height, with values > 0.3 found above 400 hPa at several sites. Indicatively, the median FGE value among the examined sites for each pressure level is > 0.2 above 400 hPa (Figure 7). As expected, the respective profiles of MNMB in Figure 6b indicate that the biases are mostly positive confirming the aforementioned discussion. The median MNMB value among the examined sites for each pressure level ranges approximately from 0 to +0.1 below 400 hPa (Figure 7), which is in agreement with the MNMB values of CAMSRA $O_3$ in the free troposphere reported by Inness et al. (2019) and Wagner et al. (2021). Above 400 hPa the respective median MNMB value ranges from +0.1 to +0.19 (Figure 7).

### 3.3 The role of chemical data assimilation

Hereafter we investigate the role of chemical data assimilation in CAMSRA $O_3$ representation during the selected STT events. To this end, we present CAMSRA no DA and observed $O_3$ profiles averaged during the selected STT events and the rest of events, at the WOUDC ozonesonde sites and IAGOS airports in Figures 8 and 9, respectively. As depicted in both figures, although CAMSRA no DA exhibits relatively higher $O_3$ concentrations during the STT events compared to the rest of events it clearly underestimates the observed $O_3$ increases in the troposphere at all sites above about 500 hPa. This, in combination with Figures 4 and 5, indicates that chemical data assimilation boosts $O_3$ concentrations in the direction of capturing the observed $O_3$ enhancement structures in the middle and upper troposphere. Similar results for the role of IFS chemical data assimilation in $O_3$ representation were reported by Akritidis et al. (2018) in their evaluation of CAMS-global forecast system during a deep STT event over Europe.

The FGE values of CAMSRA no DA $O_3$ shown in Figure 10a indicate similar values with that of CAMSRA, with an error increase close to $O_3$ enhancements in the upper troposphere. The respective MNMB values illustrated in Figure 10b reveal overall an underestimation of $O_3$ during the STT events. The median FGE and MNMB values depicted in Figure 7 suggest an overall improvement of MNMB and FGE in CAMSRA due to chemical data assimilation between 500 and 400 hPa and a deterioration above 350 hPa reflecting the aforementioned CAMSRA $O_3$ overestimation in the upper troposphere.

### 4 Conclusions

A process-oriented evaluation of CAMS reanalysis $O_3$ during tropopause folding events over the period 2003-2018 is performed using WOUDC ozonesonde data and IAGOS aircraft measurements. The selected STT events were obtained from the CAMSRA tropopause folds database by Akritidis et al. (2021) which was constructed with the implementation of the 3-D labeling and fold detection algorithm by Škerlak et al. (2015). Moreover, the role of chemical data assimilation in $O_3$ represen-

tation during the examined STT events was investigated using a CAMS control simulation without chemical data assimilation. The most notable findings of the study are summarized as follows:

- CAMSRA reproduces the observed $O_3$ increases in the troposphere during the examined folding events, which as indicated by the respective $O_{3s}$ profiles are of stratospheric origin.
- For some sites CAMSRA misses to follow the observed return of $O_3$ concentrations back to normal tropospheric levels, resulting in an overestimation of $O_3$ in the upper troposphere, with FGE values at 350 hPa ranging from 0.13 to 0.38 (median of 0.28) at the observational sites.
- The use of chemical data assimilation in IFS is found to be beneficial for the representation of CAMSRA $O_3$ enhancements in the troposphere during the STT events. However, it leads to an overestimation of $O_3$ concentrations at the upper portion of $O_3$ increases.
- Overall, and in terms of $O_3$ bias and absolute bias, only a small improvement is found between 500 and 400 hPa due to chemical data assimilation implementation.

The present analysis indicates that CAMSRA reproduces satisfactorily the observed $O_3$ increases in the troposphere during the tropopause folding events. Although IFS chemical data assimilation helps CAMSRA $O_3$ to follow the observed $O_3$ increases, it mostly leads in $O_3$ overestimation in the upper troposphere. Future improvements in the quality and vertical resolution of the assimilated $O_3$ products, increases in the vertical resolution of the IFS as well a reassessment of the $O_3$ background error statistics are expected to advance the performance of future IFS-based reanalyses in capturing $O_3$ variability during STT events.

*Data availability.* The CAMS reanalysis data are available from the CAMS Atmosphere Data Store (ADS) https://ads.atmosphere.copernicus. eu/cdsapp#!/dataset/cams-global-reanalysis-eac4?tab=form. The CAMS control simulation without data assimilation was obtained from the ECMWF's Copernicus Helpdesk. This study contains modified Copernicus Atmosphere Monitoring Service Information (2021); neither the European Commission nor ECMWF is responsible for any use that may be made of the information it contains. The ozonesonde data were downloaded from the World Ozone and Ultraviolet Radiation Data Center (WOUDC) at https://woudc.org/. The IAGOS aircraft measurements were obtained from the IAGOS database at http://iagos-data.fr/.

*Author contributions.* DA designed the study, performed the analysis, and wrote the manuscript. AP adopted the 3-D labeling algorithm and performed the fold identification calculations. JF and AI have contributed to the development of IFS and the production of CAMS reanalysis of atmospheric composition product. All authors contributed to reviewing and editing of the manuscript and interpretation of the results.

*Competing interests.* The authors declare that they have no conflict of interest.

*Acknowledgements.* This research is co-financed by Greece and the European Union (European Social Fund—ESF) through the Operational

Programme «Human Resources Development, Education and Lifelong Learning» in the context of the project "Reinforcement of Postdoctoral Researchers - 2nd Cycle" (MIS-5033021), implemented by the State Scholarships Foundation (IKY). MOZAIC/CARIBIC/IAGOS data were created with support from the European Commission, national agencies in Germany (BMBF), France (MESR), and the UK (NERC), and the IAGOS member institutions (http://www.iagos.org/partners). The participating airlines (Lufthansa, Air France, Austrian, China Airlines, Iberia, Cathay Pacific, Air Namibia, Sabena) supported IAGOS by carrying the measurement equipment free of charge since 1994. The data

are available at http://www.iagos.fr thanks to additional support from AERISM. We also acknowledge the WOUDC for making available the ozonesonde data. Michael Sprenger (ETH Zurich) is acknowledged for the original development of the 3-D labeling algorithm, which is used in the present study to identify tropopause folds. The authors gratefully acknowledge the computing facilities of the German Climate Computing Centre (Deutsches Klimarechenzentrum, DKRZ). The Copernicus Atmosphere Monitoring Service is operated by the European Centre for Medium-Range Weather Forecasts on behalf of the European Commission as part of the Copernicus program (http://copernicus.eu)

and CAMS data are freely available from atmosphere.copernicus.eu/data.

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

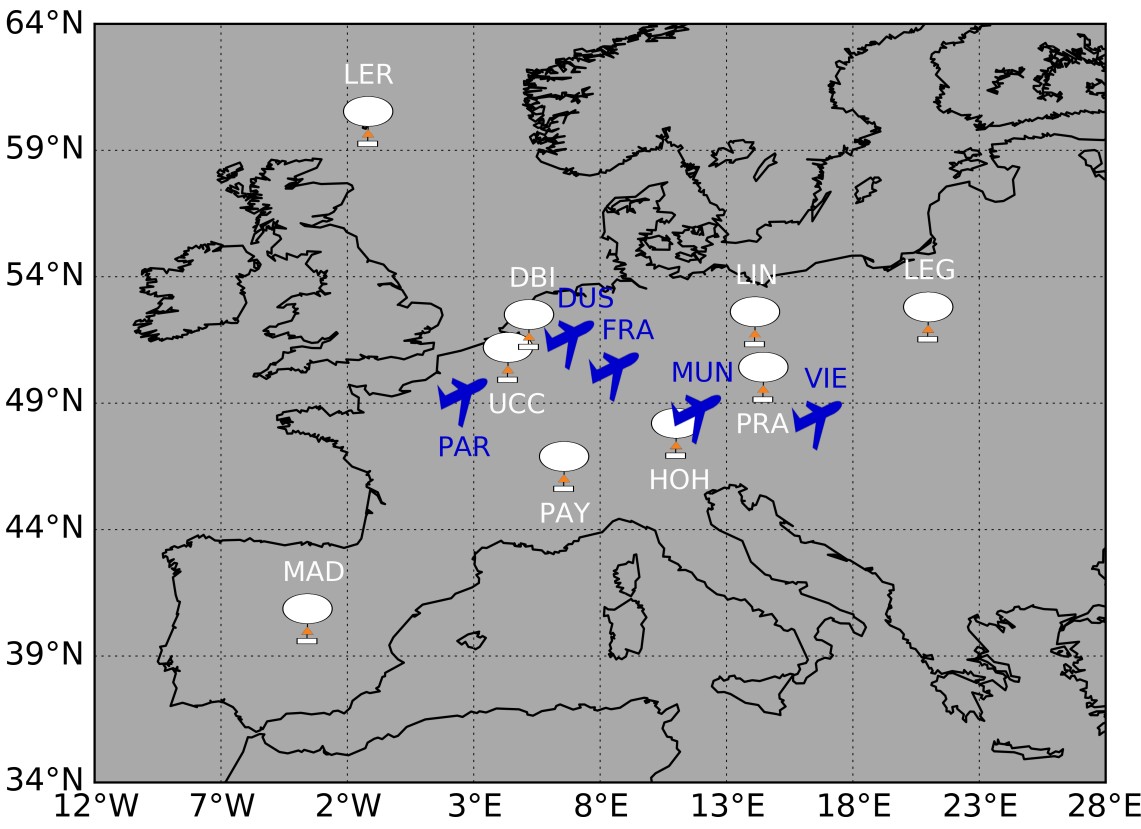

**Figure 1.** Location of WOUDC ozonesonde sites (white balloons) and IAGOS airports (blue airplanes) over Europe used in the present study.

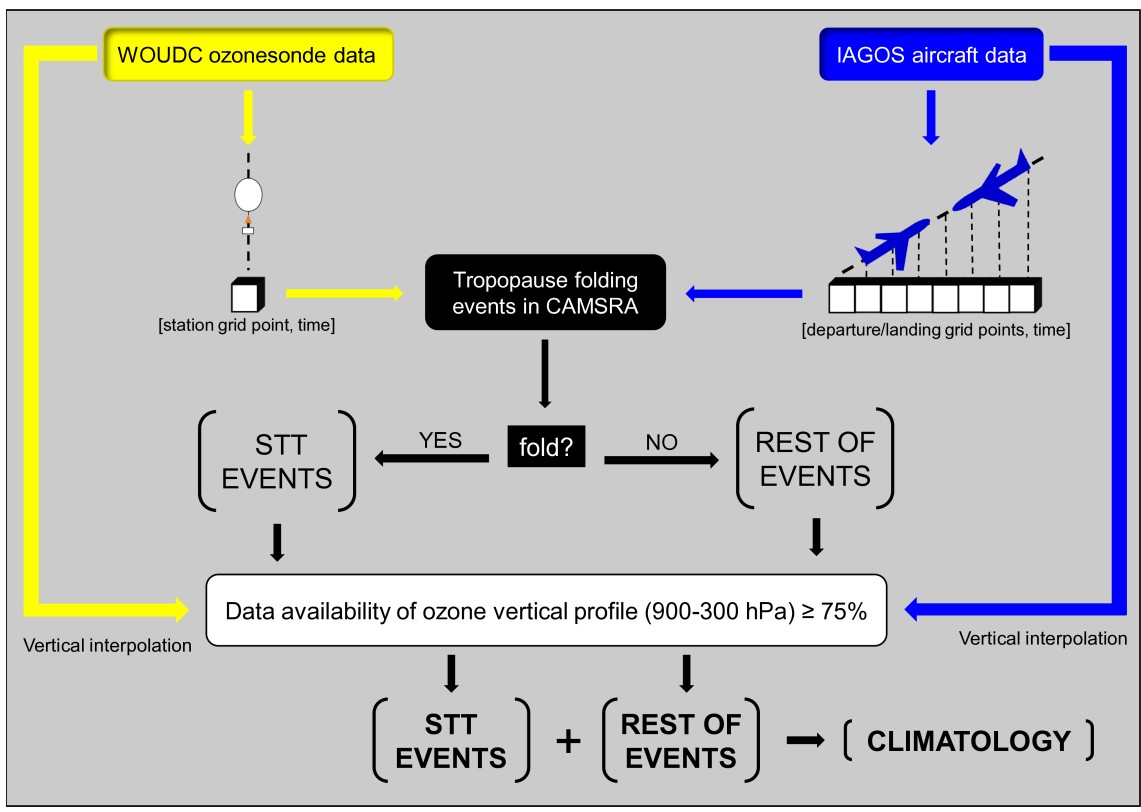

**Figure 2.** Schematic representation of the methodology applied to select STT events.

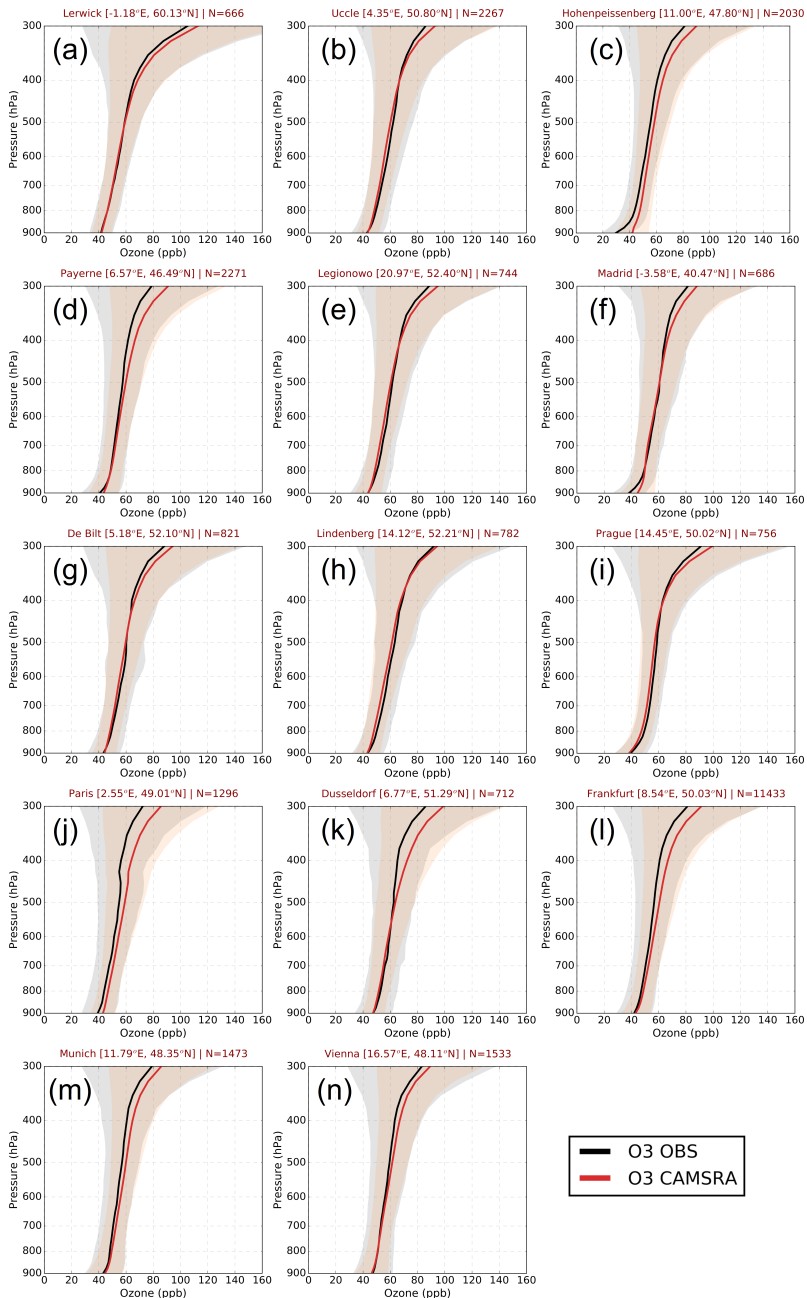

**Figure 3.** Vertical profiles of observed (black) and CAMSRA (red) ozone concentrations (ppb) at the WOUDC ozonesonde stations of a) Lerwick (UK), b) Uccle (Belgium), c) Hehenpeissenberg (Germany), d) Payerne (Switzerland), e) Legionowo (Poland), f) Madrid (Spain), g) De Bilt (the Netherlands), h) Lindenberg (Germany), and i) Prague (Czech Republic); at the IAGOS airports of j) Paris (France), k) Düsseldorf (Germany), l) Frankfurt (Germany), m) Munich (Germany), and n) Vienna (Austria) for the period 2003–2018. The grey and sandybrown shaded areas depict the ± one standard deviation of ozone vertical profiles in observations and CAMSRA, respectively.

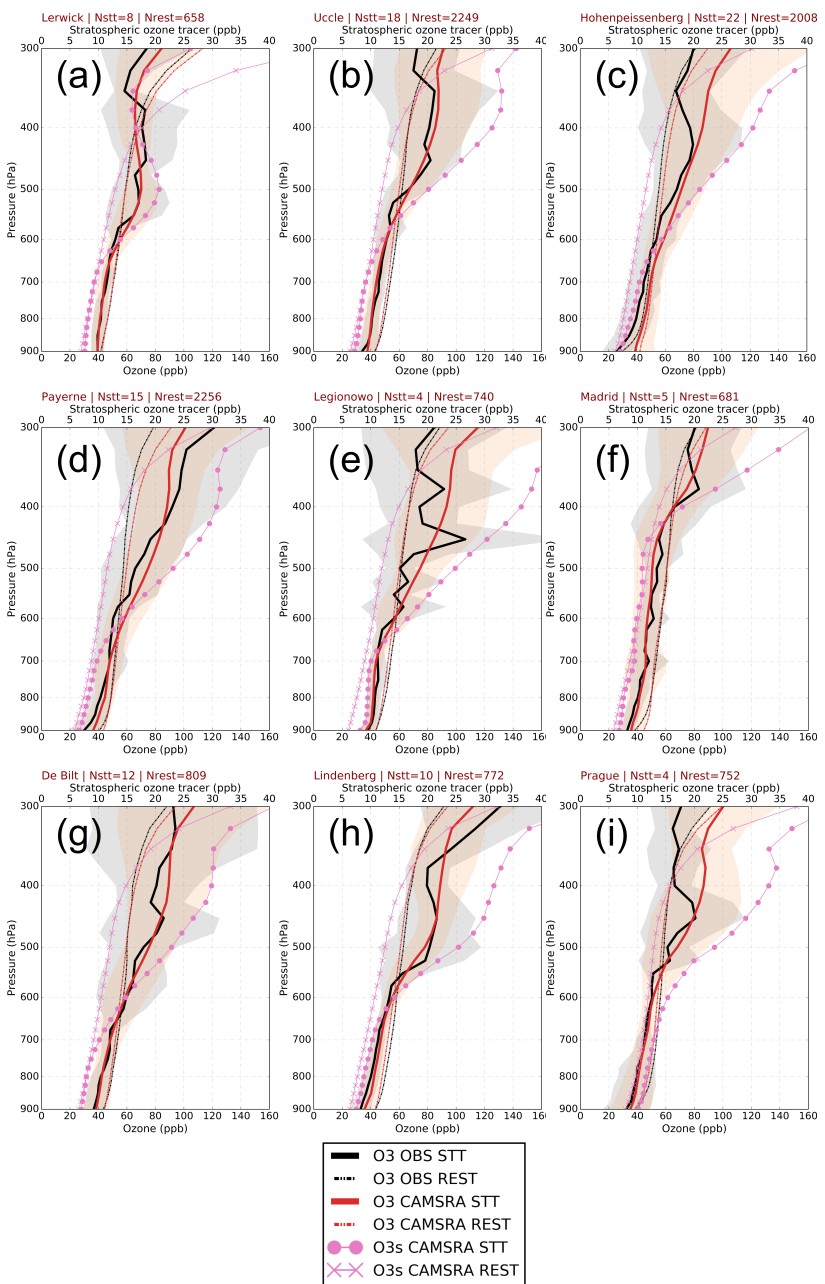

**Figure 4.** Vertical profiles of observed (black) and CAMSRA (red) ozone concentrations (ppb) during STT events (thick solid line) as well as during the rest of events (thin dashed line) at the WOUDC ozonesonde stations of a) Lerwick (UK), b) Uccle (Belgium), c) Hehenpeissenberg (Germany), d) Payerne (Switzerland), e) Legionowo (Poland), f) Madrid (Spain), g) De Bilt (the Netherlands), h) Lindenberg (Germany), and i) Prague (Czech Republic). Also shown are the vertical profiles of stratospheric ozone tracer concentrations (ppb) during STT (pink circles) and rest of events (pink x markers). The grey and sandybrown shaded areas depict the $\pm$ one standard deviation of ozone vertical profiles during STT events in observations and CAMSRA, respectively. Keep in mind that $O_3$ and $O_{3s}$ concentrations are presented in different horizontal axes.

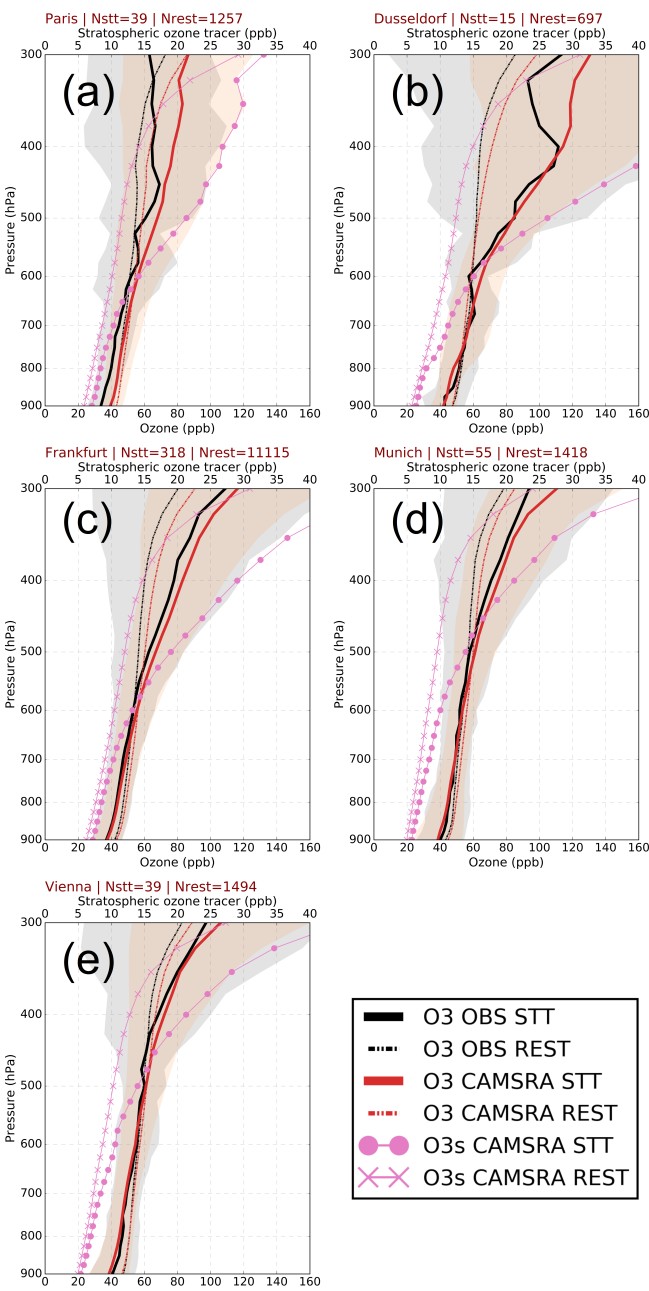

**Figure 5.** Vertical profiles of observed (black) and CAMSRA (red) ozone concentrations (ppb) during STT events (thick solid line) as well as during the rest of events (thin dashed line) at the IAGOS airports of a) Paris (France), b) Düsseldorf (Germany), c) Frankfurt (Germany), d) Munich (Germany), and e) Vienna (Austria). Also shown are the vertical profiles of stratospheric ozone tracer concentrations (ppb) during STT (pink circles) and rest of events (pink x markers). The grey and sandybrown shaded areas depict the ± one standard deviation of ozone vertical profiles during STT events in observations and CAMSRA, respectively. Keep in mind that $O_3$ and $O_{3s}$ concentrations are presented in different horizontal axes.

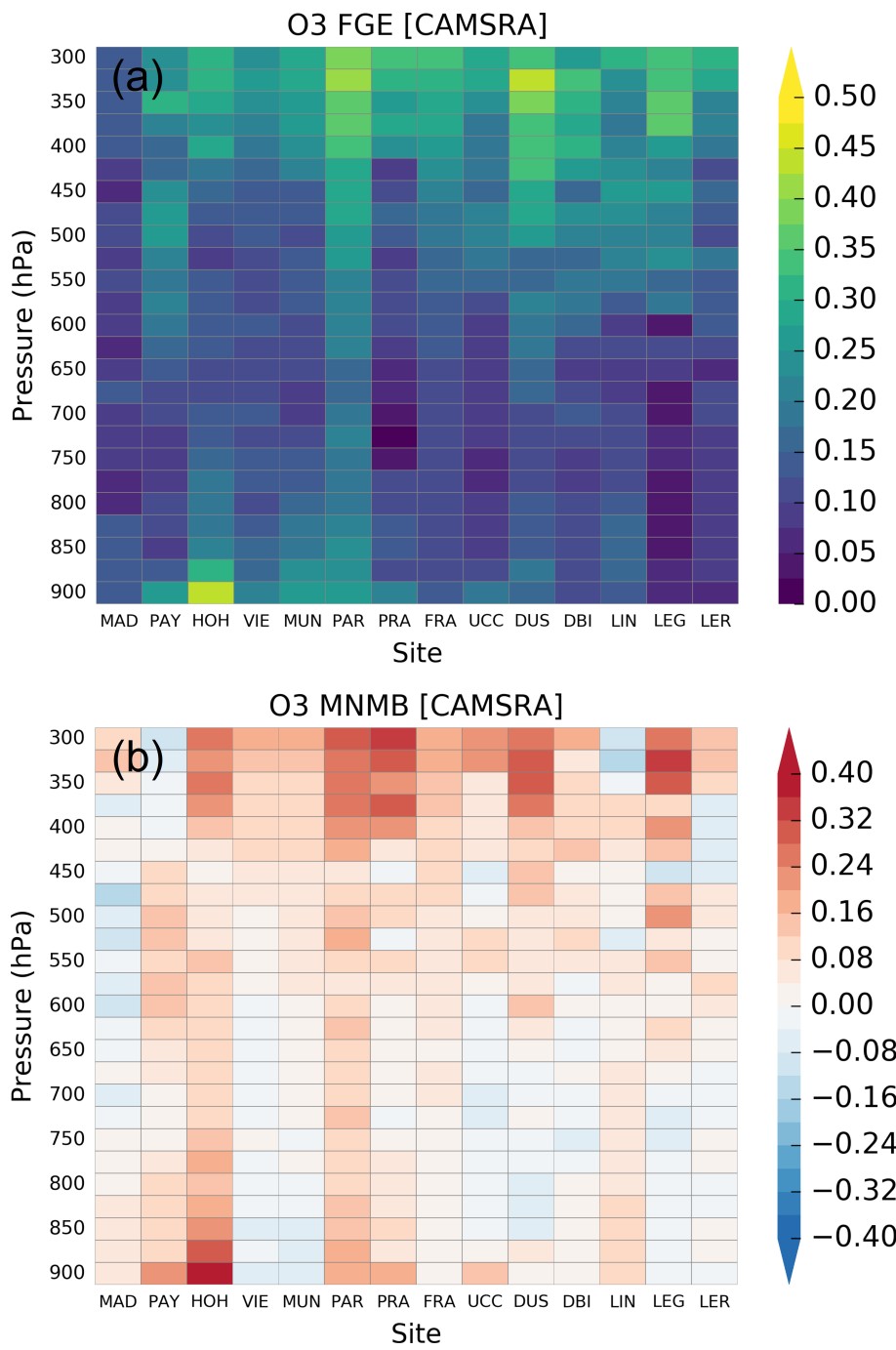

**Figure 6.** Vertical profiles of CAMSRA ozone (a) FGE and (b) MNMB for the examined WOUDC ozonesonde stations and IAGOS airports over the period 2003–2018. The observational sites are ordered with increasing latitude.

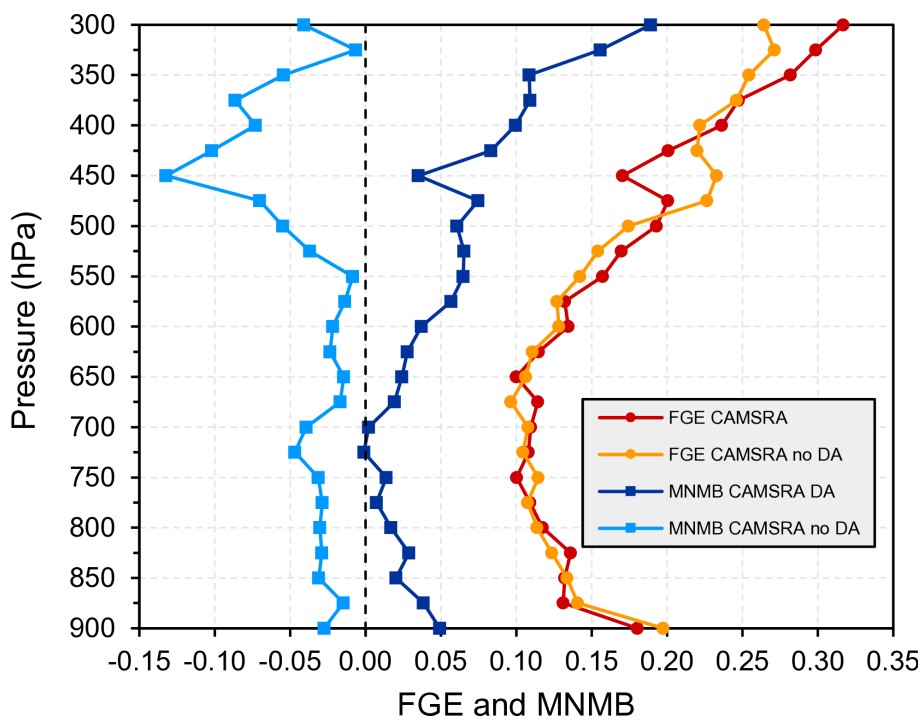

**Figure 7.** Vertical profiles of median (among examined sites) FGE and MNMB for CAMSRA O$_3$ with/without chemical data assimilation (DA/no DA).

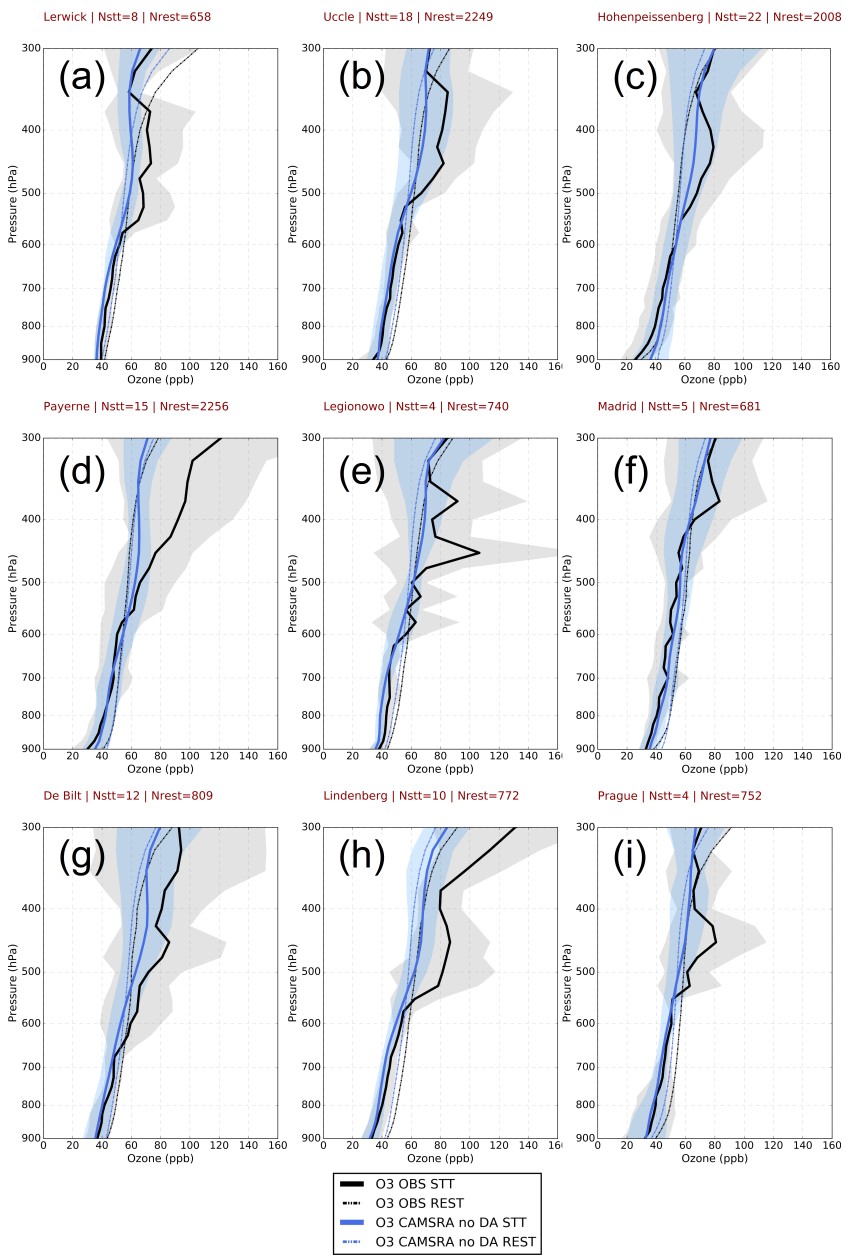

**Figure 8.** Vertical profiles of observed (black) and CAMSRA no DA (blue) ozone concentrations (ppb) during STT events (thick solid line) as well as during the rest of events (thin dashed line) at the WOUDC ozonesonde stations of a) Lerwick (UK), b) Uccle (Belgium), c) Hehenpeissenberg (Germany), d) Payerne (Switzerland), e) Legionowo (Poland), f) Madrid (Spain), g) De Bilt (the Netherlands), h) Lindenberg (Germany), and i) Prague (Czech Republic). The grey and light blue shaded areas depict the ± one standard deviation of ozone vertical profiles during STT events in observations and CAMSRA no DA, respectively.

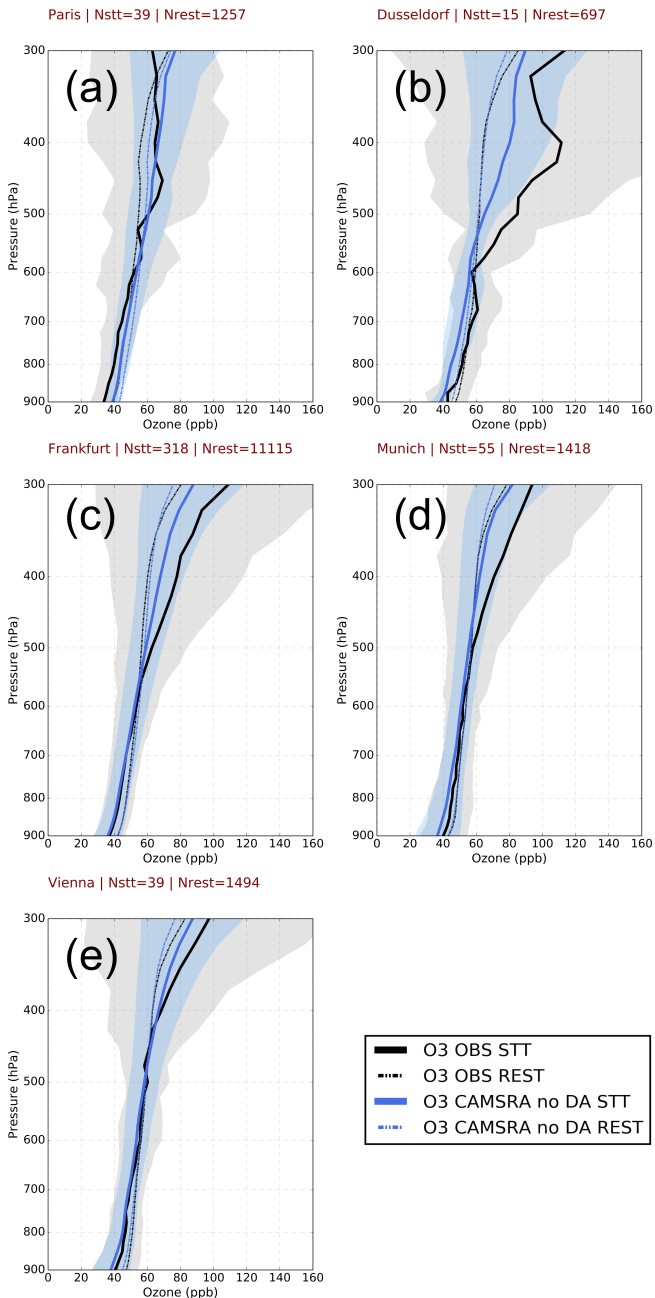

**Figure 9.** Vertical profiles of observed (black) and CAMSRA no DA (blue) ozone concentrations (ppb) during STT events (thick solid line) as well as during the rest of events (thin dashed line) at the IAGOS airports of a) Paris (France), b) Düsseldorf (Germany), c) Frankfurt (Germany), d) Munich (Germany), and e) Vienna (Austria). The grey and light blue shaded areas depict the ± one standard deviation of ozone vertical profiles during STT events in observations and CAMSRA no DA, respectively.

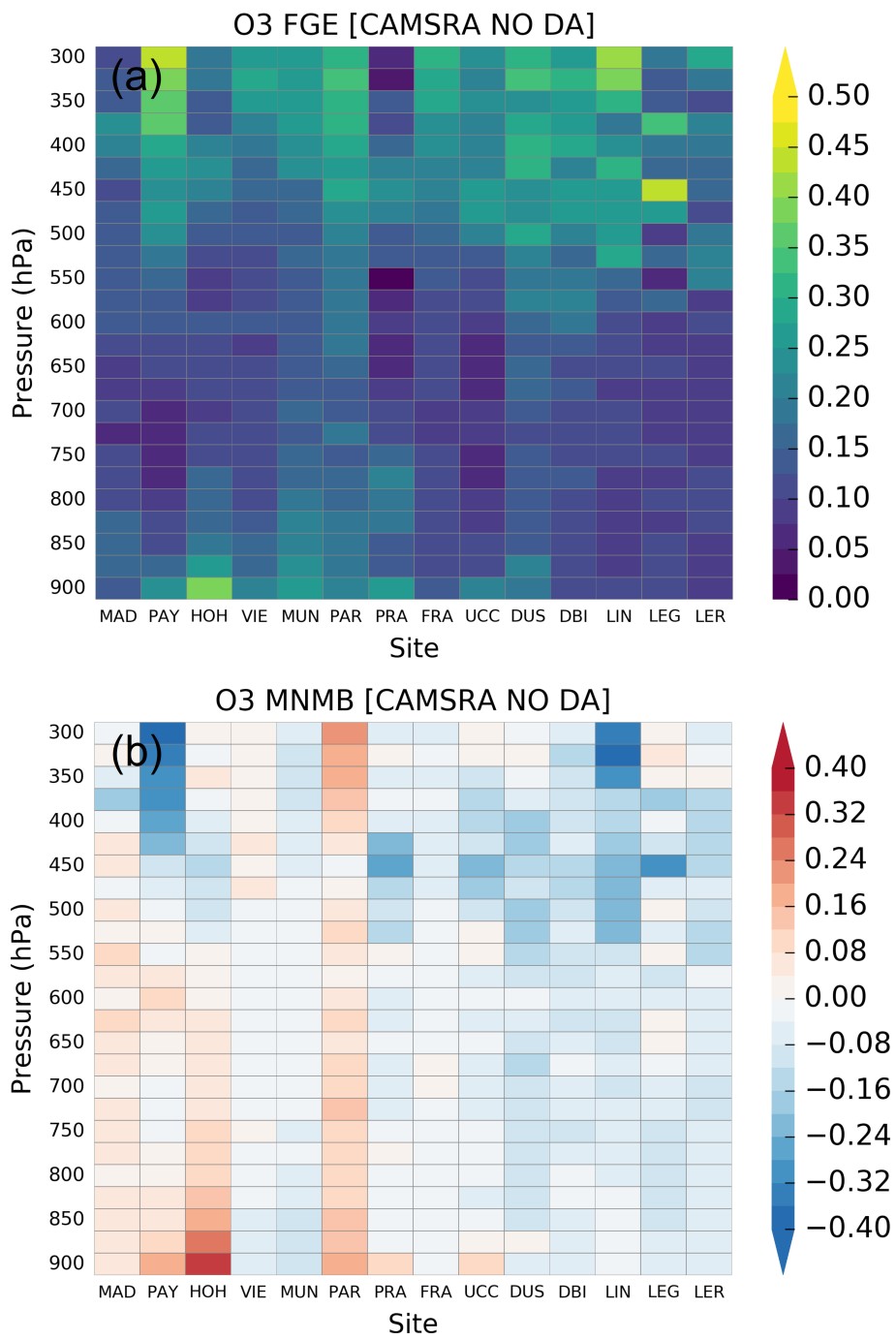

**Figure 10.** Vertical profiles of CAMSRA no DA ozone (a) FGE and (b) MNMB for the examined WOUDC ozonesonde stations and IAGOS airports over the period 2003–2018. The observational sites are ordered with increasing latitude.