# Peer review of "A process-oriented evaluation of CAMS reanalysis ozone during tropopause folds over Europe for the period 2003–2018"

_Atmospheric Chemistry and Physics, 2021_

## Author Response (AR1)

Author's Response

**"A process-oriented evaluation of CAMS reanalysis ozone during tropopause folds over Europe for the period 2003–2018"**

Dear Editor

We would like to thank the two Referees for the time devoted on reviewing our manuscript and their constructive comments which contributed to its improvement.

On the following pages we present our point-by-point response to the comments raised by the Reviewers, as well as the corresponding changes in the revised manuscript. Please mind, that these changes along with few other minor changes in the revised manuscript, can be found highlighted in the track-changes file.

Sincerely,

Dimitris Akritidis (on behalf of all the co-authors)

Note: Reviewer's comments are presented in black font; authors' responses are presented in blue plain font; manuscript text quotations are presented in blue bold font.

Anonymous Referee #1

We would like to thank Reviewer #1 for her/his time devoted and the constructive and helpful comments.

General comment:

In the manuscript, CAMS reanalysis tropospheric ozone profiles are evaluated during folding events using ozonesonde and IAGOS aircraft data in Europe. A control run without data assimilation is then also used to understand the differences, especially in the upper troposphere, between the modelled and observed ozone concentrations.

The manuscript is well written and gives a nice overview of the current knowledge about STT and tropopause folding. The scope of the manuscript is very focused, and the methodology very clear, although the knowledge of earlier studies by the authors is almost a must in the description. The results are interesting, but I had the feeling at several places that the authors could go more in depth. The authors observe, but do not interpret their findings enough. I will give examples here below.

We would like to thank Reviewer #1 for the general comments. Regarding the not so extended interpretation of some findings, we understand the rationale of the comment, yet, there is not enough flexibility in that direction as several sensitivity experiments are required which in the framework of the CAMS reanalysis are unfortunately not feasible. Nevertheless, the possible reasons for the CAMSRA O3 overestimation in the troposphere documented in the manuscript, arise from previous sensitivity experiments and experience with the IFS modeling system. More details on this and our point-by-point responses to the Reviewers comments are presented below.

- In the introduction, you might refer to the work by Zhao et al. above Asia as well https://agupubs.onlinelibrary.wiley.com/doi/10.1029/2020JD033955, https://doi.org/10.1016/j.atmosres.2020.105158, and Luo et al. (https://www.hindawi.com/journals/amete/2019/4375123/).

We have included the suggested references in the Revised Manuscript (RM).

- Why did you restrict your evaluation of the CAMS reanalysis ozone during tropopause folds to Europe? This is clearly not the region with the highest number of tropopause fold events, so I guess the availability of the ozonesonde and IAGOS profile data has driven the choice of the study area. Please clarify your choice for Europe in the introduction (lines 38-44 are not that convincing for the current focus of the manuscript).

The present work was performed within the framework of a postdoctoral fellowship of the first author by the Greek State Scholarships Foundation (IKY) (Reinforcement of Postdoctoral Re-searchers - 2nd Cycle" (MIS-5033021)) co-financed by Greece and the European Union (European Social Fund—ESF). The aim of the project is the investigation of stratospheric intrusions and their role on tropospheric ozone levels and air quality over Europe with the synergistic use of CAMS reanalysis and observational data. Thus, and as a prerequisite step in this direction, the evaluation of the CAMS reanalysis O3 during such events is limited over the European region.

- It would also be nice to give some additional climatological information on tropopause frequency, spatial and temporal variability of STT events over your study domain (Fig. 1) in Europe.

To present both the spatial and temporal variability of tropopause folds frequency over Europe we have included as a Supplement the fields of CAMSRA monthly mean folds frequency (%) for each month over the period 2003-2018. Very shallow folds with a vertical extent Δp<50 hPa are excluded. The following sentence is now included in the RM (L151-153): **"The spatial distribution of CAMSRA monthly mean tropopause folds (with $\Delta p \geq 50$ hPa) frequency over Europe for the period 2003–2018 is presented in Figure S1 of the Supplement."**

[Figure]

**Figure R1.** CAMSRA monthly mean tropopause folds (with Δp ≥ 50 hPa) frequency (%) over Europe for a) December, b) January, c) February, d) March, e) April, f) May, g) June, h) July, i) August, j) September, k) October, and l) November over the period 2003-2018.

- It is not clear to me which selection criteria have been used for the ozonesonde and IAGOS sites. For instance, the Prague ozonesonde dataset, which you have been using in an earlier study, is absent. If the data criterion for using ozonesonde time series is number of observations available throughout the 2003-2018 time period (line 71), I do not understand why, for instance, Observatoire Haute Provence (OHP), Sodankylä, Valentia (?) data were not selected. The same question arises for the IAGOS airports: what is the data temporal coverage (line 77) criterion used to include data from an airport or not?

We agree with the comment raised by the Reviewer. Initially we included the ozonesonde sites with the most available observations (in terms of number and years). Inadvertently we didn't include some sites which we do now in the RM. In the RM, the criterion used for both ozonesonde and IAGOS sites selection is to exhibit at least 500 profile observations. This subjective criterion was applied in order to ensure a sufficient number of both observational sites and folding events to be selected for analysis. As a result, except the initially examined sites two additional ozonesonde sites are included in the analysis; Lindenberg (Germany) and Prague (Czech Republic). Please mind that Figures 1, 3, 4, 6, 7, 8 and 10 are revised, yet without changing the main findings of the study. The following in now included in the RM (L92-94): **"The selection of both ozonesonde sites and IAGOS airports was based in the availability of at least 500 profile observations throughout the 2003-2018 period. This objective criterion ensures a sufficient number of both observational sites and folding events to be selected for the analysis"**.

- As you included a control run without data assimilation, please specify the sources (which satellites? which products? during which period of the 2003-2018 time frame) of partial column and profile ozone retrievals that are assimilated in CAMS reanalysis.

The partial column (PC) and profile (PR) ozone retrievals assimilated in CAMS reanalysis are presented in Table R1. This information is provided in Table 1 of the CAMS reanalysis evaluation study by Wagner et al. (2021). The following sentence is now included in the RM (L106-107): **"More details on the satellite retrievals (product, satellite, period) assimilated in CAMSRA can be found in Table 1 of the CAMSRA evaluation study by Wagner et al. (2021)."**

**Table R1.** Satellite retrievals (product, satellite, period) assimilated in CAMSRA

| Product | Instrument | Satellite | Period |
|---------|-----------|-----------|--------|
| PC 13L | SBUV/2 | NOAA-14 | 200407-200609 |
| PC 13L | SBUV/2 | NOAA-16 | 200301-200706 |
| PC 21L | SBUV/2 | NOAA-16 | 20111201-20130708 |
| PC 13L | SBUV/2 | NOAA-17 | 200301-201108 |
| PC 13L | SBUV/2 | NOAA-18 | 200507-201211 |
| PC 13L | SBUV/2 | NOAA-19 | 200903-20130708 |
| PC 21L | SBUV/2 | NOAA-19 | 20130709-20181231 |

| PR | MIPAS | ENVISAT | 20030127-20040326 20050127-20120331 |
|----|-------|---------|--------------------------------------|
| PR | MLS | AURA | 20040803-20180312 |

- In section 2.3, I would expect more details on the fold detection algorithm. Now, the summary is very limited. Major clarifications: how is the stratospheric source of air identified, what is the weight of the specific humidity content, and are the ozone concentrations used in the detection (I guess not, but please confirm clearly).

Ozone concentrations are not used as an input to the 3-D labeling algorithm, thus are not used as a proxy in fold detection. The tropopause is defined as the isosurface of PV=2 pvu or Θ=380 K, whichever is lower, so initially, an air mass is considered as stratospheric based on the criterion PV > 2 pvu or Θ > 380 K, respectively. Yet, not all masses with PV > 2 pvu should be considered to belong to the stratosphere, such as stratospheric cut-offs, surface-bound PV anomalies, and diabatically produced PV anomalies. To this end, some physical and geometrical criteria are used to categorize the air masses in five categories (5 labels): tropospheric (label=1); stratospheric (label=2); stratospheric cut-off or diabatically produced PV anomaly (label=3); tropospheric cut-off (label=4); surface-bound PV anomaly (label=5). The diabatically produced PV anomalies merged with the stratosphere are distinguished using a specific humidity threshold of 0.1 g/Kg. Further details of the applied algorithm and the criteria used for air mass labeling can be found in Škerlak et al. (2015). In the RM (L139-150) we have replaced the sentence of L113-114 with the following: **"The 3-D fields of pressure are constructed and the pressure level of the dynamical tropopause (Holton et al., 1995; Stohl et al., 2003) is determined using the lower of the isosurfaces of PV at 2 PVU and potential temperature at 380 K. Subsequently, the vertical profile for each grid point is examined and a fold is assigned when multiple crossings of the tropopause are identified. Still, there are specific cases where air with PV>2 PVU is either not connected to the stratosphere (stratospheric cut-offs) or is not of stratospheric origin (diabatic PV anomalies or surface-bound PV anomalies) which should not be considered as stratospheric. To this end, the 3-D labeling algorithm, using physical and geometrical criteria, labels the air masses as follows: tropospheric (label=1); stratospheric (label=2);**

**stratospheric cut-off or diabatically produced PV anomaly (label=3); tropospheric cut-off (label=4); surface-bound PV anomaly (label=5). The diabatically produced PV anomalies merged with the stratosphere are distinguished using a specific humidity threshold of 0.1 gkg−1. Further details on the criteria used for the 3-D labeling can be found in Škerlak et al. (2015). Therefore, a fold is identified when a 2→1→2→1 or 3 transition is detected on a vertical profile (from top to bottom), with the algorithm outputting a binary variable (0:no fold, 1:fold) for every grid point and time step."**

- The selection of STT events (section 2.4) in ozonesonde and IAGOS profile data seems to be rather indirect , based on your database of STT events detected in CAMSRA (with 3D-labeling algorithm). I assume this algorithm is not directly applicable to ozonesonde and IAGOS "2D" data? Please specify. However, algorithms exist to detect tropopause folds in ozonesounding data as well (e.g. Van Haver et al., https://doi.org/10.1029/96GL00956), so how did you confirm the presence of an STT event independently from CAMSRA in the ozonesonde and IAGOS data? Details are missing how a fold is found in the ozonesonde profile (line 126) and in the IAGOS profiles (lines 131-135).

No, the algorithm is not applicable to ozonesonde and IAGOS data. The comparison between CAMSRA and observations indicates that during the selected folds the observed ozone exhibits a clear increase compared to the rest of events confirming that in principle folds are also present in observations. Furthermore, the visual inspection of the individual IAGOS vertical ozone profiles for the 318 selected STT (fold) events over FRA indicates that in ~93% of the profiles clear ozone increases were seen in some part of the troposphere. In the rest of the profiles, the increase might be small in both observations and CAMSRA, due to small impact of the specific folds or issues related to temporal and horizontal resolution of CAMSRA.

 - At the end of section 2.4, it would be nice to include some statistical information: how many of the CAMSRA STT events are also identified in the coincident ozonesonde and IAGOS profiles? Are the detection rates site dependent? Dataset (ozonesonde vs. IAGOS) dependent? What are the relative ratios between the STT events and rest of events? Again: site or dataset specific? Consistent?

Based on our methodology the fold events are identified in CAMSRA only, so such statistical information is not feasible. An explicit detection of folds from the observational data would rather serve as an evaluation of the 3-D labeling algorithm which is beyond the scope of this study. Moreover, a methodology for folds detection based on observed ozone concentrations would be not directly comparable to the 3-D labeling algorithm detection of folds which is based on CAMSRA meteorological data.

- In section 3.1 and 3.2, you mention some spatial differences between the CAMSRA and observational ozone profile differences, e.g. in lines 145-148, in lines 157-159. But, you do not give any explanation why features in the differences arise at some sites, and not at other sites. Is this related to data quality issues at some sites, instruments used, differences in the spatial representativeness of the tropospheric ozone observations at some sites compared to others, etc? Some discussion and/or thoughts would be helpful here.

Regarding the climatological comparison between CAMSRA and observations, all stations exhibit an overestimation of CAMSRA O3 in the upper troposphere which is the main feature. Attributing the different overestimations among the examined sites in specific reasons is difficult, but potential factors can indeed be discussed. To our knowledge no site-specific data quality issues are reported, but both ecc and Brewer Mast ozonesonde measurements introduce uncertainties which might affect in different ways the comparison among the examined sites. In addition, the proximity of the selected grid points to the respective ozonesonde site, and the CAMSRA 3-D spatial representation of the IAGOS take-off/landing are also another factor to consider. Accordingly, we have included the following sentence in the RM (L184-187): **"The differences seen in the comparison between the observed and CAMSRA O3 concentrations among the examined sites are subject to the uncertainties introduced by the ozonesonde instrument measurements, as well as the proximity of the selected grid points to the respective ozonesonde sites, and the CAMSRA 3-D spatiotemporal representation of the IAGOS take-off/landing routes."**

As for the differences of CAMSRA-obs comparison during the selected STT events, in addition to the aforementioned factors and even more importantly is the vertical location and geometrical characteristics of these events. More

specifically, the pressure level in which a part of the fold appears over the examined site, directly affects ozone concentrations, and shapes its vertical profile. Thus, for observational sites with not so extended number of stt events the individual dynamics and ozone increases in different levels of the troposphere can form somehow unique structures of CAMSRA and observed O3 deviations. In contrast, sites where more stt events were selected (e.g. Frankfurt and Munich) exhibit more smoothed profiles for both CAMSRA and observations.

[Figure]

**Figure R2.** Schematic representation of fold detection.

In more detail, Figure R2 presents a schematic representation of fold detection, also depicting the multiple crossings of the stratosphere in vertical profiles. As mentioned in the manuscript the pressure difference between the middle and upper stratosphere crossing $\Delta p = p_m - p_u$ indicates the vertical extent of the fold, while the vertical area between the $p_m$ and $p_l$ represents the area of the fold that directly affects the ozone profile over the underlying site. Such crossings during the stt events detected at the ozonesonde sites (not shown for IAGOS as several folds might be present during a selected event following the take-off/landing route) are depicted in Figure R3. The

following sentence is now included in the RM (L205-207): **"In addition, the individual dynamics and the different vertical location and geometrical characteristics of the selected STT events, especially for observational sites with not so extended number of events, may form somehow unique structures of CAMSRA and observed O3 deviations"**.

[Figure]

**Figure R3.** Vertical location of middle (blue) and lower (red) crossings of the stratosphere during STT events at the WOUDC ozonesonde sites.

- My previous comment was on the spatial fingerprint of the CAMSRA ozone evaluation. But what about the temporal fingerprint? Is there a temporal evolution in the figures 3 to 10 that is smeared out by considering only the full 2003-2018 period? There might be a temporal component due to a change of the data quality of the observations, change of data assimilation source data, etc. Can you comment if you detected a temporal fingerprint?

We thank the Reviewer for the comment. A potential factor affecting upper tropospheric ozone concentrations in CAMSRA is suggested to be biased assimilated data. In particular, it seems that when Aura (MLS V4 data) start to be assimilated in August 2004 an overestimation of CAMSRA ozone in the upper troposphere arises that is not seen in the control simulation (CAMSRA no DA) (Figure R4). This is also supported if we average all examined sites and split the comparison between CAMSRA and observations by year, for both stt and rest of events, as depicted in Figures R5 and R6, respectively.

During 2003 and 2004 there is a better agreement (with even a small underestimation) in the upper troposphere between CAMSRA and observations suggesting that the inclusion of Aura data in 2004 is probably a driver of the upper tropospheric ozone overestimation. The following is now included in the RM (L202-205): **"In particular, CAMSRA O3 vertical profiles during both STT and rest of events exhibit a better agreement in the upper troposphere with observations during the years 2003 and 2004, indicating that the inclusion of the Aura data in the assimilation system from August 2004 and on is likely to result in O3 overestimation in the upper troposphere (Figures S2 and S3 in the Supplement).".**

[Figure]

**Figure R4** Timeseries of CAMSRA no DA (left) and CAMSRA DA (right) O3 relative differences (%) against 13 European ozonesondes (40-61N, 10W-24E).

[Figure]

**Figure R5.** Vertical profiles of observed (black) and CAMSRA (red) ozone concentrations (ppb) averaged over all examined WOUDC and IAGOS sites during STT events for each year of the period 2003-2018. The grey and sandybrown shaded areas depict the ± one standard deviation of ozone vertical profiles during STT events in observations and CAMSRA, respectively.

[Figure]

**Figure R6.** Vertical profiles of observed (black) and CAMSRA (red) ozone concentrations (ppb) averaged over all examined WOUDC and IAGOS sites during the rest of events (no STT) for each year of the period 2003-2018. The grey and sandybrown shaded areas depict the ± one standard deviation of ozone vertical profiles during rest of events in observations and CAMSRA, respectively.

- There is no clear explanation given in section 3.3 why chemical data assimilation deteriorates the comparison with the observations above 350 hPa. Only in the conclusions, a list of possible improvements in the data assimilation is given, which might explain the larger inconsistencies between model and observations at those pressure levels.

As also stated in our previous responses sensitivity simulations to isolate the impact of potential drivers of this overestimation are not feasible at the time in the framework of CAMS reanalysis. This deterioration reflects the CAMSRA O3 overestimation in the upper troposphere discussed in previous comments, thus in the RM we have included the following phrase at the end of the last sentence in section 3.3 (L241): **"..reflecting the aforementioned CAMSRA O3 overestimation in the upper troposphere."**

- Please reformulate the expression "with a bias increase close to O3 increases closes to the upper troposphere"; its meaning is not fully clear to me.

The phrase has been replaced (L237-238) with **"with a bias increase close to O3 enhancements in the upper troposphe".**


We would like to thank Reviewer #2 for the general comments. Our point-by-point responses to the Reviewers comments are presented below.

Major comments:

- Please be more specific about the locations of folding hot spots. Why you are especially interested over Europe? I suggest to add some additional references on the global STT over different hot spots of fold activities.

Similar to our response to a comment raised by Reviewer #1, the present work was performed within the framework of a postdoctoral fellowship of the first author by the Greek State Scholarships Foundation (IKY) (Reinforcement of Postdoctoral Re-searchers - 2nd Cycle" (MIS-5033021)) co-financed by Greece and the European Union (European Social Fund—ESF). The aim of the project is the investigation of stratospheric intrusions and their role on tropospheric ozone levels and air quality over Europe with the synergistic use of CAMS reanalysis and observational data. Thus, and as a prerequisite step in this direction, the evaluation of the CAMS reanalysis O3 during such events is limited over the European region.

In the Revised Manuscript (RM) we have included additional references for the global STT over different hot spots of fold activity. The following are now included in the RM (L40-L44): **"The springtime western United States region is a hot spot of deep folding events with well-known implications for tropospheric ozone and air quality (Langford et al., 2009; Lin et al., 2012, 2015; Knowland et al., 2017). Recently, Luo et al. (2019) explored the seasonal features of tropopause folds over the Tibetan Plateau where folds occur frequently (Tyrlis et al., 2014), while other studies investigated the effect of tropopause folds on lower tropospheric ozone levels and air quality in China (Lu et al., 2019; Zhao et al., 2021b, a)."**

- How did you perform the control simulation of IFS without the use of chemical data assimilation? Since you claimed that the chemical data assimilation is the key factor resulting in most of the biases, you need to be more specific about your experiment design.

The following is now included in the RM as a description for the control run (L120-126): **"As it would have been computationally too expensive to produce a control analysis experiment that was identical to CAMSRA but did not actively assimilate observations of reactive gases, a forecast run was carried out that applied the same settings (model code, resolution, emissions) as used in CAMSRA. The control run was carried out as a sequence of 24 hours. The meteorological initial conditions were taken from CAMSRA, but the initial conditions for the atmospheric composition species, including ozone, from the previous forecast. It thus allows us to detect the impact of the assimilation of e.g. ozone data by comparing its ozone fields with CAMSRA."**

- I have concerns about the great differences between O3 and O3s in the upper troposphere (300-400hPa) as shown in your Figure 4 and 5. Green profiles differ largely away from the red ones at 300 hPa. Will they merge near the tropopause level? If not, I am very worried about the quality of the upper tropospheric O3 as well as the stratospheric ozone tracer in CAMSRA. can you please show a horizontal map of both the climatologies of O3 and O3s at 100 hPa and 850 hPa, respectively? Additionally, there have been various methods when calculating the O3s tracer among different models. Please add more details about the stratospheric ozone tracer (O3s) in CAMSRA. What is the choice of tropopause in defining the tropopause? I'm

curious about it because the diagnostics of STT might be sensitive to the choice of tropopause definition.

We thank the Reviewer for this comment. Indeed, O3 and O3s concentrations differ between 300 and 400 hPa. As initially stated in the manuscript O3s, the stratospheric ozone tracer is identical with ozone in the stratosphere. This was the intension but unfortunately it turns out that this is not the case, due to an issue in the coupling of O3s to O3 in the stratosphere. So CAMSRA O3s in the stratosphere is only the modeled (Cariolle) ozone as data assimilation were not applied for that. As suggested by the Reviewer, Figure R1 presents the CAMSRA 2003-2018 climatology of O3 (left) and O3s (right) concentrations at 100 hPa (top) and 850 hPa (bottom). The abovementioned issue is reflected on the differences between O3 and O3s at 100 hPa, as well as on the vertical profiles of O3 and O3s concentrations over the 2003-2018 period (Figure R2). Tropopause in CAMSRA is calculated based on the temperature lapse rate, switching the chemistry scheme from CB05 (troposphere) to CARIOLLE (stratosphere) accordingly.

It must be noted, that in the present study O3s is used only for qualitative purposes to support the fact that during the selected STT events the induced CAMSRA O3 increase is associated with stratospheric O3 transport. Thus, we do not refer to specific O3s amounts as of transported to the troposphere anywhere in the manuscript. So, the fact that O3s is not generated from the O3 fields after assimilation is not crucial for the purpose of O3s usage here. In the RM manuscript (L129-134) we have replaced the O3s description with the following: **"Apart from O3, a stratospheric ozone tracer (O3s) is also used from CAMSRA providing a diagnostic of O3 STT. In principal, O3s in IFS is defined identically with O3 in the stratosphere, yet, in CAMSRA O3s is equal to the modeled (Cariolle scheme) O3 tracer and not the assimilation-resulted O3. In the troposphere O3s is subject to transport and chemical destruction just like O3. The tropopause in CAMSRA is calculated based on the temperature lapse rate, switching the chemistry scheme from CB05 (troposphere) to CARIOLLE (stratosphere) accordingly. It should be noted, that O3s is used here only as a qualitative diagnostic of ozone STT, to support evidence of stratospheric ozone downward transport during the folding events."**

[Figure]

**Figure R1.** CAMSRA 2003-2018 climatology of O3 (left) and O3s (right) concentrations at 100 hPa (top) and 850 hPa (bottom).

[Figure]

**Figure R2.** Vertical profiles of CAMSRA 2003-2018 climatology of O3 (black) and O3s (red) concentrations. The vertical axis displays CAMSRA model levels (from surface to the top).

- Please add more details about the two evaluation metrics (FGE vs MNMB) chosen in your study. Why you chose these two statistics and what are differences between the two? What does the score mean, respectively?

Since O3 concentrations are in general increased with altitude exhibiting higher values near the tropopause, we prefer to show normalized metrics for easier and more straightforward interpretation. The fractional gross error (FGE) is a normalized version of the mean error metric, while the modified normalized mean bias (MNMB) is a normalized version of the mean bias. Both metrics are normalized by the mean of the observed and CAMSRA values and have the advantage to behave symmetrically with respect to under- and overestimation, being less sensitive to outliers in the distribution. They are dimensionless and relative, making them suitable for comparison at different parts of the troposphere. The FGE is a measure of the overall error of CAMSRA, while MNMB is a measure of the overall bias. The MNMB metric is useful in the direction of showing if and how much overall CAMSRA under- or overestimates the observed O3 concentrations, while the FGE is used to depict the overall CAMSRA error regardless under- or overprediction. Both metrics are widely used in atmospheric composition evaluation studies related to CAMS (e.g. Katragkou et al., 2015; Akritidis et al., 2018; Inness et al., 2019; Wagner et al., 2021). In the RM we have modified the respective paragraph as follows (L210-215): **"For a quantitative comparison between CAMSRA and observations, we present in Figures 6 the vertical profiles of fractional gross error (FGE) and modified normalized mean bias (MNMB) of CAMSRA O3 for the WOUDC ozonesonde sites and IAGOS airports. The FGE is a normalized version of the mean error, while the MNMB is a normalized version of the mean bias. Both metrics are normalized by the mean of the observed and model (here CAMSRA) value, being dimensionless and relative, thus suitable to use at different heights in the troposphere. FGE and MNMB are insensitive to outliers in the distribution, and range between 0 to 2 and -2 to 2, respectively, behaving symmetrically with respect to under- and overestimation."**

- Section 3.1 and 3.2, what leads to the different biases across stations?

Similar to our response in a comment raised by Reviewer #1: Regarding the climatological comparison between CAMSRA and observations, all stations

exhibit an overestimation of CAMSRA O3 in the upper troposphere which is the main feature. Attributing the different overestimations among the examined sites in specific reasons is difficult, but potential factors can indeed be discussed. To our knowledge no site-specific data quality issues are reported, but both ecc and Brewer Mast ozonesonde measurements introduce uncertainties which might affect in different ways the comparison among the examined sites. In addition, the proximity of the selected grid points to the respective ozonesonde site, and the CAMSRA 3-D spatial representation of the IAGOS take-off/landing are also another factor to consider. Accordingly, we have included the following sentence in the RM (L184-187): **"The differences seen in the comparison between the observed and CAMSRA O3 concentrations among the examined sites are subject to the uncertainties introduced by the ozonesonde instrument measurements, as well as the proximity of the selected grid points to the respective ozonesonde sites, and the CAMSRA 3-D spatiotemporal representation of the IAGOS take-off/landing routes."**

As for the differences of CAMSRA-obs comparison during the selected STT events, in addition to the aforementioned factors and even more importantly is the vertical location and geometrical characteristics of these events. More specifically, the pressure level in which a part of the fold appears over the examined site, directly affects ozone concentrations, and shapes its vertical profile. Thus, for observational sites with not so extended number of stt events the individual dynamics and ozone increases in different levels of the troposphere can form somehow unique structures of CAMSRA and observed O3 deviations. In contrast, sites where more stt events were selected (e.g. Frankfurt and Munich) exhibit more smoothed profiles for both CAMSRA and observations.

In more detail, Figure R3 presents a schematic representation of fold detection, also depicting the multiple crossings of the stratosphere in vertical profiles. As mentioned in the manuscript the pressure difference between the middle and upper stratosphere crossing $\Delta p = p_m - p_u$ indicates the vertical extent of the fold, while the vertical area between the $p_m$ and $p_l$ represents the area of the fold that directly affects the ozone profile over the underlying site. Such crossings during the stt events detected at the ozonesonde sites (not shown for IAGOS as several folds might be present during a selected

[Figure]

**Figure R3.** Schematic representation of fold detection.

[Figure]

**Figure R4.** Vertical location of middle (blue) and lower (red) crossings of the stratosphere during STT events at the WOUDC ozonesonde sites.

event following the take-off/landing route) are depicted in Figure R4. The following sentence is now included in the RM (L205-207): **"In addition, the individual dynamics and the different vertical location and geometrical characteristics of the selected STT events, especially for observational sites with not so extended number of events, may form somehow unique structures of CAMSRA and observed O3 deviations".**

Minor comments:

- Lines 30-35: please be more specific about the hot spots of fold activities. Please add some references for transport of VSLS to the lower stratosphere

The hot spots of fold activity are presented in the next paragraph (see L38-50 in the RM). Regarding the transport of VSLS to the lower stratosphere the following phrase was included in the RM (L27-29): **"The latter constitutes an important pathway through which very short lived substances (VSLS), emitted at the surface, can be transported to the lower stratosphere influencing ozone (Levine et al., 2007; Aschmann et al., 2009; Liang et al., 2014)."**

- Line 39, a comma is missing before "resulting"

Done.

Line 68: can you add a few sentences about the differences between the ecc and the Brewer Mast ozonesondes?

The following is now included in the RM (L75-78): **"Both ozonesonde types are based on the same measurement principle of ozone electrochemical detection in potassium iodine. The major differences between ecc and Brewer Mast ozonesondes are that the latter uses only one reaction chamber, and a silver anode instead of a platinum anode, requiring an external electrical potential in contrast to the ecc (Beekmann et al., 1994)."** Moreover, the Komhyr (1969) and Brewer and Milford (1960) references are now included in the RM.

Line 90: can you be more specific about "CY42R1" and "4D-VAR"?

Further details on CAMS reanalysis are provided at https://confluence.ecmwf.int/display/CKB/CAMS%3A+Reanalysis+data+documentation. Although documentation for CY42R1 is not available, details on earlier and later IFS cycles for reference can be found at https://www.ecmwf.int/en/publications/ifs-documentation. In the RM we have included further information on the 4D-VAR system and the meteorological observations assimilated in IFS: (L102-104) **"In more detail, it is based on the minimization of a penalty function that takes the**

**deviations of the model's background fields from the observations to provide the optimal forecast during 12-hour assimilation windows (from 09 UTC to 21 UTC and 21 UTC to 09 UTC) by modifying accordingly the initial conditions."** and (L107-109) **"In addition, meteorological observations, including satellite, PILOT, in situ, radiosonde, dropsonde, and aircraft measurements are also incorporated in IFS."**.

- Line 92: $NO_2$

Done.

- Line 99: what is the vertical resolution in the upper troposphere?

There are 13 levels between approximately 400 and 100 hPa. The following is now included in the RM (L115): **"(13 levels between approximately 400 and 100 hPa)"**.

- Line 130: the rest of

Done.

- Line 142: I suggest adding the longitude and latitude information of each station to Figure 3.

Done.

- Line 166: Figure

Done.

---

## Referee Report (RR1)

**Review of "A process-oriented evaluation of CAMS reanalysis ozone during tropopause folds over Europe for the period 2003-2018" by Akritidis et al.**

Thank you for carefully answering and taking into account most of my comments. The figures that have been included in the supplementary material are really an added value for the paper and some figures (figures R1 and R4/R5) might even deserve to be included in the main manuscript, I would say. The authors have now provided the essential information about the methods and tools used in their analysis and the findings are also much more interpreted in the revised version of the manuscript.

I only have two minor comments:

1. The reason you gave in your response why you concentrate on Europe (project related) is not very scientific. Please include some scientific arguments (can also be data availability) why the region of interest was Europe, and not other regions in the world with more SST or tropopause fold events.

2. I think you could do better in explaining the spatial variability of the O3S/IAGOS-CAMSRA ozone differences: "The differences seen in the comparison between the observed and CAMSRA O3 concentrations among the examined sites are subject to the uncertainties introduced by the ozonesonde instrument measurements, as well as the proximity of the selected grid points to the respective ozonesonde sites, and the CAMSRA 3-D spatiotemporal representation of the IAGOS take-off landing routes. " As all the considered sites use ECC sondes (it should also be written in capital letters in the manuscript, not ecc), except Hohenpeissenberg, the ECC ozonesonde uncertainties should be rather modest and very similar for the different ECC sites (so no explanation for the site to site variability). BM sondes experience a higher challenge for measuring tropospheric ozone, but, on the other hand, the Hohenpeissenberg people have a long experience with it. Also the IAGOS instruments at the different airports should be traceable to the same standard, so this cannot explain why the Paris observed profiles deviate much more from CAMSRA than the ones at other airports. The CAMSRA model output should give you an idea about the spatio-temporal variability of tropospheric ozone around the sites/airports: is this higher around Hohenpeissenberg and Paris compared to the other sites? In this context, how are the sites ordered in Fig. 6 and Fig. 10? Making a geographical ordering (e.g. increasing latitude or longitude) might make sense for those figures.

---

## Author Response (AR2)

Author's Response

**"A process-oriented evaluation of CAMS reanalysis ozone during tropopause folds over Europe for the period 2003-2018"**

Dear Editor

Please find below our responses to the Reviewer's #2 comments.

In addition, to make our Figures more color-blind friendly (suggestion from the editorial support), we have modified Figure 3 (quality of red color), Figures 4 and 5 (quality of red color, green is replaced with pink), and Figures 6 and 10 (new color schemes).

Sincerely,

Dimitris Akritidis (on behalf of all the co-authors)

Note: Reviewer's comments are presented in black font; authors' responses are presented in blue plain font; manuscript text quotations are presented in blue bold font.

**Anonymous Referee #2**

We would like to thank Reviewer #2 for her/his time devoted and the constructive and helpful comments.

Thank you for carefully answering and taking into account most of my comments. The figures that have been included in the supplementary material are really an added value for the paper and some figures (figures R1 and R4/R5) might even deserve to be included in the main manuscript, I would say. The authors have now provided the essential information about the methods and tools used in their analysis and the findings are also much more interpreted in the revised version of the manuscript.

We would like to thank Reviewer #2 for the positive comments. Our point-by-point responses to the Reviewers comments are presented below.

I only have two minor comments:

1. The reason you gave in your response why you concentrate on Europe (project related) is not very scientific. Please include some scientific arguments (can also be data availability) why the region of interest was Europe, and not other regions in the world with more SST or tropopause fold events.

Since for the European region the observational data exhibit relatively higher data availability (following our selection criteria) compared to other regions around the world, we have included the following sentence in the Revised Manuscript (RM) (L63-64): **"Compared with other regions worldwide, the European region exhibits relatively higher observational data availability for the examined period."**

2. I think you could do better in explaining the spatial variability of the O3S/IAGOS-CAMSRA ozone differences: "The differences seen in the comparison between the observed and CAMSRA O3 concentrations among the examined sites are subject to the uncertainties introduced by the ozonesonde instrument measurements, as well as the proximity of the selected grid points to the

respective ozonesonde sites, and the CAMSRA 3-D spatiotemporal representation of the IAGOS take-off landing routes. " As all the considered sites use ECC sondes (it should also be written in capital letters in the manuscript, not ecc), except Hohenpeissenberg, the ECC ozonesonde uncertainties should be rather modest and very similar for the different ECC sites (so no explanation for the site to site variability). BM sondes experience a higher challenge for measuring tropospheric ozone, but, on the other hand, the Hohenpeissenberg people have a long experience with it. Also the IAGOS instruments at the different airports should be traceable to the same standard, so this cannot explain why the Paris observed profiles deviate much more from CAMSRA than the ones at other airports. The CAMSRA model output should give you an idea about the spatio-temporal variability of tropospheric ozone around the sites/airports: is this higher around Hohenpeissenberg and Paris compared to the other sites? In this context, how are the sites ordered in Fig. 6 and Fig. 10? Making a geographical ordering (e.g. increasing latitude or longitude) might make sense for those figures.

The annual mean (2003-2018) CAMSRA O3 concentrations over the examined sites indicate small spatial variability. In more detail, the CAMSRA O3 concentrations at Hohenpeissenberg and Paris are not higher than that of the other sites, but rather similar, as also indicated by Figure 3. As discussed in the paper the CAMSRA 03 overestimation in the upper troposphere is a feature seen in majority of examined stations. The overestimation the seen at Hohenpeissenberg and Paris in the middle and lower troposphere is on average ~3 ppb (between 450 and 850 hPa) which is slight amount comparable with the precision of sonde measurements and within the range of the standard deviations in ozone profiles for observations and CAMSRA. Similar O3 overestimations in the free troposphere over Hohenpeissenberg are also reported for previous ECMWF atmospheric composition reanalysis products such as the MACC reanalysis (see Figures 7 and 8 in the evaluation study by Katragkou et al. (2015)).

Moreover, CAMSRA O3 representation is also subject to regional differences and uncertainties in ozone precursor emissions affecting modeled local net photochemical ozone production rates. In addition, differences seen in the comparison between the observed and CAMSRA O3 concentrations among the examined sites are also related to the spatiotemporal representativeness of WOUDC vertical profiles and IAGOS aircraft take-off/landing routes by the selected CAMSRA grid points and time steps. In the RM we have replaced the respective sentence with the following (L185-189): **"The differences seen in the comparison between the observed and CAMSRA O3 concentrations among the examined sites are presumably related to regional differences and**  uncertainties in O3 precursor emissions affecting modeled local net photochemical O3 production rates, as well as the spatiotemporal representativeness of WOUDC vertical profiles and IAGOS aircraft takeoff/landing routes by the selected CAMSRA grid points and time steps."

Finally, as suggested, in Figures 6 and 10 in the RM the sites are ordered with increasing latitude, and "ecc" is replaced with "ECC".

**References**

Katragkou, E., Zanis, P., Tsikerdekis, A., Kapsomenakis, J., Melas, D., Eskes, H., Flemming, J., Huijnen, V., Inness, A., Schultz, M. G., Stein, O., and Zerefos, C. S.: Evaluation of near-surface ozone over Europe from the MACC reanalysis, Geosci. Model Dev., 8, 2299–2314, https://doi.org/10.5194/gmd-8-2299-2015, 2015